# Effects of Cardiac Rehabilitation in Cardiopulmonary Fitness with High-Risk Myocardial Infarction

**DOI:** 10.3390/healthcare10101849

**Published:** 2022-09-23

**Authors:** Seok Yeon Choi, Ji Hee Kim

**Affiliations:** Department of Rehabilitation Medicine, School of Medicine, Wonkwang University, 895, Muwang-ro, Iksan 54538, Korea

**Keywords:** myocardial infarction, cardiac rehabilitation, cardiopulmonary function

## Abstract

The prevalence of acute coronary syndrome in Korea has steadily increased, however, the understanding of and participation rate in cardiac rehabilitation (CR) is very low. There are few studies have been conducted in myocardial infarction (MI) patients with reduced heart function, a so-called high-risk group. Therefore, it is necessary to compare the effects of CR on the degree of improvement in cardiopulmonary fitness (CPF), whether MI patients participate or not, especially in the patients that are at a high risk of MI. Three hundred and ninety-four patients that were commissioned for CR between January 2016 and December 2020 were screened for risk stratification based on the American Association of Cardiovascular and Pulmonary Rehabilitation guidelines, and 115 were classified as high-risk patients. We retrospectively reviewed the patients who underwent both an exercise tolerance test (ETT) during the initial visit and 3 months after the onset of the study. During this period, 42 subjects were included, of which, 26 underwent at least one CR session and 16 did not. The baseline characteristics of the patients showed no significant differences. The results of the CPF improvement were measured as peak oxygen consumption (VO_2 peak_) and metabolic equivalent of tasks (METs) values which were derived through the ETT. Prior to the ETT, all of the demographic features, including ejection fraction, showed that there were no significant differences between the two groups. The initial CPF values were the same. However, after three months, the VO_2 peak_ and METs values showed that there were significant differences between the two groups (*p* < 0.01). Additionally, the exercise time differed significantly between the two groups. The CPF values and exercise time showed a significant increase after 3 months in the CR participants. Therefore, it is necessary to initiate cardiac rehabilitation especially in high-risk patients as soon as the patient’s vital signs are stable to improve their cardiopulmonary function.

## 1. Introduction

The prevalence of acute coronary syndrome (ACS), which is defined as a suspicion or confirmation of acute myocardial ischemia or infarction (MI), in Korea has steadily increased since 2015 by 17.2% (13.8 million), from 80.4 million in 2015 to 94.2 million in 2019. Despite the rapid development of the treatment for ACS, the increased risk of sudden cardiac death (SCD) and the occurrence of heart failure (HF) after treatment lowers the quality of life and survival rate [1,2]. In particular, it has been reported that the lower the left ventricular ejection fraction is (LVEF) (especially below 35%), then the higher the probability of a SCD occurring 2 years after the onset of ACS [3]. Since the treatment of ACS only focuses on the acute stage of the disease, the understanding of and participation rate in cardiac rehabilitation (CR) is very low [4]. For example, CR has been covered by health insurance since 2017 but only 1.5% of patients have participated in Korea [5]. Several factors that lower the participation rate in CR include the limited number of treatment institutions, comorbidities or functional decline, distance and the financial burden [6]. Therefore, studies on the safety and effectiveness of CR for high-risk patients who have a higher occurrence of complications are lacking. CR plays a very important role in improving clinical outcomes [7,8]. Recent studies have shown that patients who participated in CR reduced the 5-year mortality rate by 59% [9]. Despite the necessity of CR in high-risk patients, participation is limited due to safety concerns of exercise or comorbidities. Even though CR is important in MI patients, most prior studies have investigated CR in relation to chronic heart failure. There are no studies that have investigated the efficacy of CR especially in high-risk patients who are prone to complications. This may be due to the patients’ unstable vital signs, early complications or the fear of relapse. Therefore, this study aimed to investigate “How effective CR is especially for reduced heart function, in high-risk patients compared with those that did not participate in CR”.

## 2. Materials and Methods

### 2.1. Subjects

We retrospectively analysed data from 394 patients that were commissioned for CR at the University Hospital between January 2016 and December 2020. The patients were screened for a risk stratification based on the American Association of Cardiovascular and Pulmonary Rehabilitation (AACVPR) program; patients were considered to be at high risk if any of the following factors were present:LVEF < 40%;A survivor of cardiac arrest or sudden death;Having complex ventricular dysrhythmias (ventricular tachycardia, frequent [>6/min] multiform premature ventricular contractions) at rest or during exercise;Having undergone MI or cardiac surgery that was complicated by cardiogenic shock, congestive heart failure, and/or signs or symptoms of post-procedure ischemia;Having an abnormal hemodynamic profile with exercise, especially flat or decreasing systolic blood pressure or chronotropic incompetence with increasing workload;Having significant silent ischemia (ST depression ≥ 2 mm without symptoms) with exercise or recovery;Having signs/symptoms including angina pectoris, dizziness, light headedness, or dyspnoea at low levels of exercise (<5.0 METs) or in recovery;Having a maximal functional capacity that is less than 5.0 METs;Having clinically significant depression or depressive symptoms.

Patients who attended at least one session of CR were included in the ‘CR participants’ group, but patients who did not participate in any CR session were placed in the ‘CR non-participants’ group. We excluded the patients with the following criteria: an inability to perform CR, a history of congenital heart disease or valvular heart disease, cerebrovascular disease, and musculoskeletal pain. This study was conducted in accordance with the Declaration of Helsinki. All of the experimental protocols were approved by the Institutional Review Board of the University Hospital committee, and the participants signed informed consent forms for their participation.

### 2.2. Data Abstraction

Complementary clinical data and other variables were manually collected from the University Hospital electronic medical records by the authors. The health records included demographics, such as age, sex, left ventricular ejection fraction (LVEF), coronary heart disease (CHD) event type, body mass index (BMI), and other comorbidities (e.g., hypertension, diabetes, dyslipidemia, smoking status, etc.). The LVEF was determined using transthoracic echocardiography, which was performed within 1 week of the onset of this study.

### 2.3. Exercise Tolerance Test

All patients who were commissioned for CR underwent an exercise tolerance test (ETT) to measure their exercise capacity. We used a treadmill (Q-stress TM55, Mortara Instrument, Inc., Milwaukee, WI, USA), and their autonomic blood pressure and heart rate was recorded (247BP, SunTech Medical, Morrisville, NC, USA) using a metabolic gas analyzer (TrueOne 2400, ParvoMedics, Inc., USA). A 12-lead electrocardiogram (Quinton Q-stress, Mortara Instrument, Inc., USA) was attached when the patients were at rest and their cardiopulmonary exercise responses were recorded during exercise and after the completion of the test to prevent any complications such as cardiac arrest or arrhythmia, etc. To evaluate the patients’ cardiopulmonary fitness (CPF), all of the subjects completed a symptom-limited exercise test using the modified Bruce protocol according to the AACVPR guidelines by the supervising clinician.

The following indices were measured: maximal systolic and diastolic blood pressure (SBP _max_, DBP _max_), maximal heart rate (HR _max_), peak oxygen consumption (VO_2 peak_), volume of air exchanged per minute (VE _peak_), metabolic equivalent of tasks (METs), respiratory exchange ratio (RER), the final stage of the ETT and the duration of exercise were measured in patients who completed the ETT. The VO_2 peak_ is a parameter that reflects the highest rate of oxygen uptake and utilization by the body during intense activity. METs represents a simple procedure for expressing the energy cost of physical activities as multiples of the resting metabolic rate. It is a commonly used method to quantify the physical activity level or work output. These two parameters (VO_2 peak_ and METs) are the most valuable indices that reflect a patient’s CPF.

Based on the results of the initial ETT, the intensity of the exercise was gradually adjusted from 40% to 85% by calculating the spare heart rate based on the resting heart rate and the maximal heart rate of each subject which was obtained through the initial ETT. The CR program consisted of 1 h sessions that were conducted 1 or 2 times a week for a total of 12 weeks. Each session consisted of a 10 min warm-up, a 40 min main exercise, and a 10 min cool-down. The main exercise consisted of 30~40 min of treadmill activity. Three months after the event, their exercise capacity was evaluated again, and the time of the onset of this study was compared in both of the CR participants and non-participants that were using the ETT.

### 2.4. Echocardiography

Doppler-echocardiography was performed to measure the left ventricular ejection fraction (LVEF) and end-systolic and end-diastolic diameters (LVESD and LVEDD) between the patients during the initial test and the test that took place 3 months after the event. The parasternal long-axis, short-axis at the papillary muscle level, and apical 4- and 2-chamber views were recorded. The left ventricular ejection fraction (LVEF) and end-systolic and end-diastolic diameters (LVESD and LVEDD) were measured according to the Simpsons model. The Doppler-echocardiographic studies were all performed by the same cardiologist who was blinded to the study.

### 2.5. Statistical Analysis

The Mann–Whitney U test was used to compare the ages and diagnoses of the two groups. A Pearson’s chi-square test and Fisher’s exact test were used to compare sex, smoking status, and risk factors for CHD. A Student’s *t*-test was used to compare the changes in VO_2 peak_, VE _peak_, RER, final stage ETT, METs, exercise time, SBP _max_, DBP _max_, and HR _max_ between the two groups to compare their exercise capacity. A paired *t*-test was used to compare the ETT indices at the time of the onset of the study and 3 months later. SPSS ver. 28.0 (IBMSPSS, Armonk, NY, USA) was used for all of the statistical analyses. The statistical significance was defined as *p* < 0.05.

## 3. Results

### Demographic Characteristics

The demographics and characteristics of the patients are shown in Figure 1 and Figure 2. A total of 115 high-risk patients were included from 2016 to 2020 according to the AACVPR risk stratification at the baseline. Of those high-risk patients, 65 patients did not complete the initial test or the ETT that occurred 3 months later due to a follow-up loss, three patients did not perform the ETT that occurred 3 months later at the appropriate period (5 or 6 months after the onset), four patients did not satisfactorily perform the exercises (RER less than 0.9), and one patient could not successfully perform the ETT due to recent knee pain. Thus, 42 patients were assessed, including eight post-cardiopulmonary resuscitation patients. A total of 26 patients (61.9%) participated in the CR program and 16 patients (30.1%) did not.

The average patient age in the CR and CR non-participant groups was 61.3 and 64.3 years, respectively, showing that there was no significant difference in sex distribution. The ejection fraction was 39.3% in CR participants and 35.3% in CR non-participants, also showing that there was no significant difference between the two groups. Furthermore, there were no significant differences in other demographic features, including type of CHD event, hypertension, diabetes mellitus, dyslipidemia, smoking status, and BMI. (Table 1).

At the initial ETT, all of the exercise capacity indices including VO_2 peak_, VE _peak_, RER, the final stage of ETT, the METs, and the exercise time did not show any statistically significant difference (Table 2).

After the ETT that occurred 3 months after the first one, the VO_2 peak_ increased from an average of 23.07 to 26.24 mL/kg/min in the CR participant group and from 19.71 to 21.45 mL/kg/min in the CR non-participant group. These results show that there is a statistically significant difference between the CR participants and the non-participants 3 months after the onset of the study (*p* = 0.03). Additionally, there were significant differences in the final stages of the ETT, the METs, and the exercise time. The average final stage of the ETT increased from 4.81 to 4.88 in the CR participants; while in the CR nonparticipant group, it decreased from 4.19 to 3.38 (*p* < 0.01). The average of the METs increased from 6.60 to 7.49 in the CR participants and from 5.64 to 6.13 in the CR non-participants (*p* = 0.04) groups. The average exercise time increased from 753.38 to 768.08 s in the CR participant group; however, in CR non-participants group, the exercise time decreased unexpectedly from 685.50 to 500.13 s (*p* < 0.01). The average VE _peak_ increased from 59.71 to 70.02 L/min in the CR participant group and from 52.53 to 56.15 L/min in the CR non-participant group. Contrary to our expectations, the VE _peak_ showed was no statistically significant difference in the ETT that occurred 3 months after the first one (*p* = 0.06). Although the RER, SBP _max_, DBP _max_, and the maximal heart rate showed an increasing trend in both groups, there were no significant differences in the results of the ETT that occurred 3 months after the first one (*p* > 0.05).

No statistically significant differences were observed in the respiratory exchange ratio and SBP _max_, and DBP _max_ in either the CR participants or non-participants groups. The change in VO_2 peak_, VE _peak_, and the METs increased significantly after the ETT that occurred 3 months after the first one in the CR participants group only (*p* < 0.05). However, the final stage of the ETT and the exercise time showed a statistically significant decrease in only the CR non-participants group. Additionally, the maximal heart rate showed an increase in the results of the second ETT in both groups (*p* < 0.05).

According to the results in Table 2 and Table 3, the average VO_2 peak_ and the METs in the ETT that occurred 3 months later showed a statistically significant difference between the two groups, but the amount of change only showed a significant increase in the CR participants group. In contrast to the previous results, the average final stage of the ETT and the exercise time also showed a statistically significant difference between the two groups, but the amount of change only showed a significant decrease in the CR non-participants group. While the amount of change in VE _peak_ showed a significant increase in both the CR participants and non-participants group, the average of both of these indices showed no statistically significant difference in the results of the ETT the occurred 3 months later.

In the CR participants group, the LVEF increased from 37.76 ± 8.68% to 46.77 ± 11.82% (change rate 9.01 ± 11.28%), and in the CR non-participants group, the LVEF increased from 36.24 ± 4.03% to 43.64 ± 8.74 (change rate 7.41 ± 8.21%). The LVEF results were statistically significant in both groups. The absolute change value of the LVEF (CR participants, 9.01 ± 11.28%; CR non-participants, 7.41 ± 8.21%) seemed to be similar in both groups, but as the two groups had different baseline LVEF values, we also calculated the relative change rate in addition to the absolute change. However, those results did not show any significant difference in both groups (Table 4). Also, no significant changes were observed in LVEDD and LVESD, regardless of the CR (*p* > 0.05).

## 4. Discussion

The present study compared the exercise capacities in the case of myocardial infarction with high-risk groups between patients with and without CR training. After 3 months, the VO_2 peak_ and METs values showed a significant increase in the high-risk patients who participated in the CR programs (*p* < 0.01). In case of the VE _peak_, it did not show the same results; however, when this was compared within each group, the CR participants showed a significant change, whereas the CR non-participants did not, suggesting that CR affects the VE _peak_ enhancement (*p* < 0.05). These results support the importance of initiating CR in high-risk patients.

It is important to optimize the proper exercise intensity to maximize a patient’s heart function recovery [10]. However, there can be always possible complications, such as arrythmia, which can be fatal. This study was conducted in patients with an LVEF that was less than 40% or those who experienced cardiac arrest, which was among the indicators corresponding to high-risk patients. Therefore, to perform the CR safely, the patients were stratified by the AACVPR guidelines to prevent problems that may have arisen from exercise before the start of the program. Before starting the CR, the ETT was preceded by the prescription of appropriate exercise intensity according to individual results, and this was performed safely in all of the patients. The intensity of aerobic exercise in the CR provides recommendations for the ECG monitoring during the patient’s exercise period. Careful monitoring was performed by experienced nurses and physical therapists during the exercise training period of the CR, with well-prescribed exercise that was of appropriate intensity for the patient’s condition. With respect to the exercise test, none of the subjects demonstrated ST segment changes that suggested cardiac ischemia or arrhythmias during exercise. All of the patients demonstrated a normal hemodynamic and electrocardiography response to the exercise test and an improved exercise capacity and LVEF.

Patients with high-risk acute MI have a higher incidence of HF, which leads to increased mortality [11]. They also have a higher risk of sudden death after acute MI. Low LV function is a predictor of mortality with a 20–40% risk of sudden cardiac death within 1 year of acute MI [12]. CR after MI is highly recommended [13] in guidelines (class IA), as it has been proven to be effective; however, its participation rate is still low [14]. The hesitation to participate in high-risk patients is due to the fear of exercise-related complications, the deterioration of the patient’s existing condition, or accompanying depression. Although there was no statistical significance in our study, the mean value of the VO_2 max_ reflecting the initial LVEF and CPF was lower in the CR group. This indicates that patients with serious illnesses should participate more in CR, even though they experience difficulty in participating.

Exercise has been reported to have a good effect on the prognosis of patients with ischemic heart disease [15]. It has been reported that exercise attenuates abnormal remodeling in patients with chronic HF [16], and myocardial fibrosis in patients with MI, thereby preserving cardiac function. A meta-analysis on the effect of exercise reported a significant effect in reducing LVESD and increasing VO_2 peak_, which had a good effect on improving the left ventricular remodeling in patients with LV dysfunction after MI [17]. In patients with chronic HF that was induced by ischemic heart disease, the application of an exercise program for 6 months improved their exercise capacity [18]. In our study, the CPF values showed significant improvement only in the CR group. These results prove that CR is a very effective and essential treatment, even in high-risk patients.

In addition to the exercise parameters, several biomarkers are being studied to represent the cardiac function. L-arginine, a substrate that is used by nitric oxide (NO) synthase, has shown to have beneficial effects on driving endothelial vasodilatation, reducing inflammation, and ameliorating physical function. One study shows the result that of the oral supplementation of L-arginine induces the response to CR after MI and cardiac revascularization [19]. Another study showed that the vasopressin surrogate marker copeptin (CPP) improved significantly after CR. Moreover, improved CPP was correlated with peak oxygen uptake and exercise intensity, which are two of the most important indicators of cardiorespiratory capacities [10].

High-risk patients are more likely to develop HF in the future. Even HF patients with better physical skills and functions have low mortality and hospitalization rates, regardless of their LV function [20]. In healthy adults, each 1-MET increase in CPF is associated with a 15% decrease in mortality [21]; therefore, improving CPF is important to improve the patients’ clinical outcomes. Consequently, functional training may effectively increase the HF patient’s cardiorespiratory capacities and their quality of life [22]. Moreover, a multicenter retrospective cohort study also reported that the participation in CR by patients with HF had a significant benefit in improving their prognosis regardless of their age, sex or comorbidities [23].

In this study, the prevalence of diabetes mellitus was at 33.2% among the subjects. Exercise in CR enhances the health outcomes of patients with type 2 DM by improving their endothelial function, inflammation [24], and hyperactivated sympathetic system [25,26]. Endothelial dysfunction that is associated with DM is an important risk factor for acute MI, and MI is a major cause of death in patients with DM. DM with a history of MI has a recurrence of MI of over 40% [27], and one of the risk factors for the progression of HF in patients with acute MI is DM [28]. Therefore, the stricter control of DM is needed to reduce the risk of the recurrence of MI [29]. Other studies have reported that the participation rate of CR in patients with DM was lower than that of non-DM patients, and that CR reduced the mortality rate in DM patients [30].

In our study, only the CR-participants group showed a significant improvement in the CPF, such as the METs; however, both groups showed a statistically significant enhancement in their ejection fraction 1 year later. In contrast, there were no significant differences between the results of the initial period and those taken year later in all of the indices, including the EF (Table 4).

Additionally, the author also confirmed that other echocardiographic parameters, LVESD and LVEDD, showed no significant difference in either group (Figure 3 and Table 4), which correlates with the results of previous studies in high-risk patients [31]. One study that was conducted in Japanese patients revealed that an increased epicardial adipose tissue volume may affect the hemodynamics and CPF by decreasing the peak oxygen uptake [32]. This result implies that when one is performing echocardiography in MI patients, the epicardial volume should also be measured.

The limitations of this study include this being a retrospective study and having data that were collected at one center. Moreover, few patients received the CR as many patient dropped out after the screening. There were 115 high-risk patients but for several reasons, 72 patients dropped out. The most common reason was due to a loss in the follow-up visits. This correlates with the reality of a low participation rate in Korea, despite the patients being at high risk. More studies with a large sample size are needed to further explain the effects of CR. Previous studies were limited to comparing the treatment effect between high-risk and low-risk patients, but this study only targeted high-risk patients and showed there is a meaningful difference in improving exercise capacity depending on patient participation in CR. The general effects of CR, such as increased physical activity, smoking cessation, education on healthy food habits, and stress management programs, are thought to have the same effects in the high-risk group. This comprehensive CR program has positive effects on blood sugar and pressure management, smoking cessation, and drug compliance.

After the COVID-19 pandemic, many patients became sedentary, and this reduced the participation rate in the CR. One of the solutions to combat this problem was to transfer the existing CR program in an institute setting to a home-based telerehabilitation. Marie et al. reported that a 3-week home-based telerehabilitation was effective and improved the patients’ cardiopulmonary fitness, thereby representing a safe alternative CR method [33]. This type of approach to CR may facilitate the continuity of care for patients and increase the participation rate.

## 5. Conclusions

Cardiac rehabilitation is safe and necessary even in high-risk patients, despite their low participation rate. Cardiopulmonary fitness increased in the high-risk patients that participated in the cardiac rehabilitation compared to those who did not. Although cardiac rehabilitation has a significant effect on improving cardiopulmonary fitness in MI patients, high-risk patients still do not participate in CR easily due to the higher incidence of complications. However, this study showed there were no serious complications when cardiac rehabilitation was performed under supervision. Therefore, it should be strongly recommended that patients initiate this treatment at the same time as medications for secondary prophylaxis in order to address the low CR participation rate. However, there is a need to conduct more studies with a larger sample size to further explain the effects of cardiac rehabilitation in high-risk groups.

## Figures and Tables

**Figure 1 healthcare-10-01849-f001:**
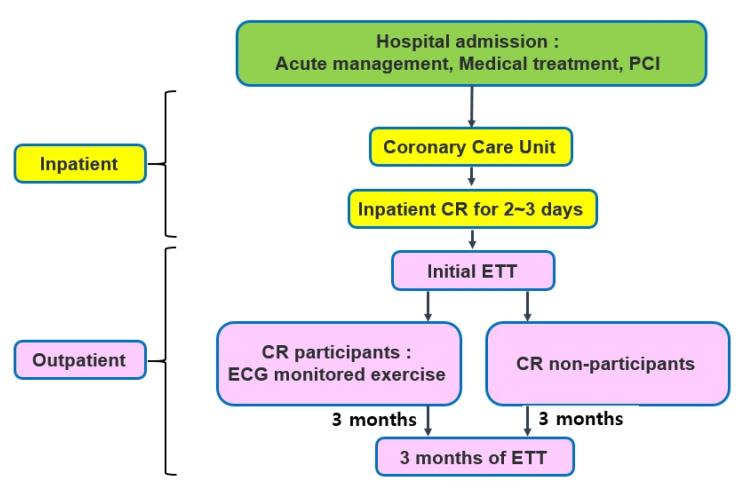
Study design. Abbreviations: CR, cardiac rehabilitation; PCI, percutaneous coronary intervention; ETT, exercise tolerance test; ECG, electrocardiography.

**Figure 2 healthcare-10-01849-f002:**
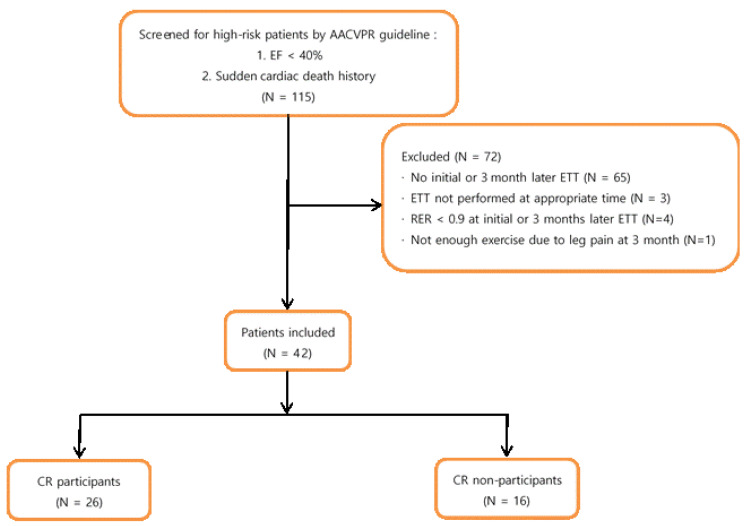
Data extraction process.

**Figure 3 healthcare-10-01849-f003:**
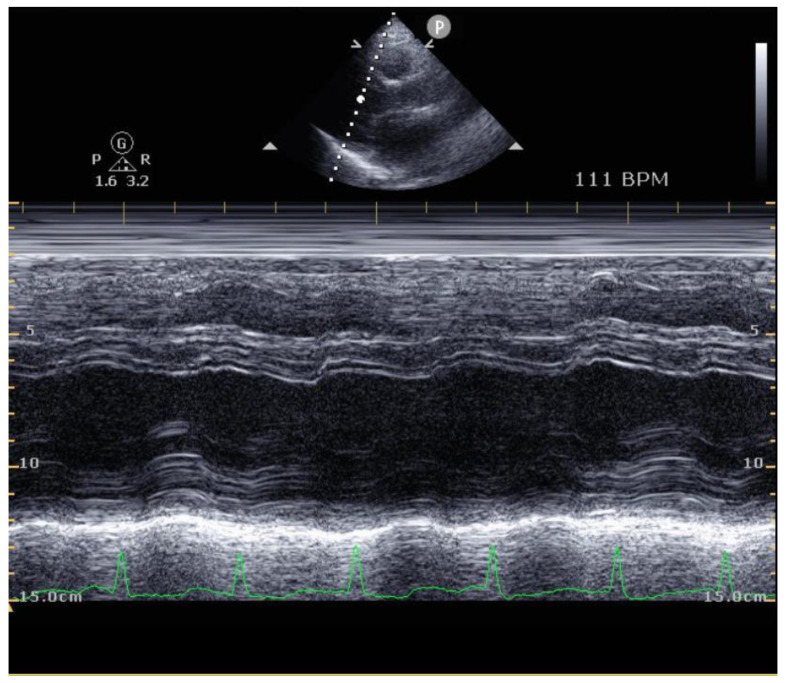
LVEDD and LVESD measured with the M-mode on parasternal long axis view. Abbreviations: LVESD, left ventricular end-systolic diameters; LVEDD, left ventricular end-diastolic diameters.

**Table 1 healthcare-10-01849-t001:** Clinical demographic characteristics of the subjects.

		Cardiac Rehabilitation Participants (*N* = 26)	Cardiac Rehabilitation Non-Participants (*N* = 16)	*p*-Value
Age (years)		61.3 ± 6.52	64.3 ± 8.70	0.213
Sex ratio (men/women)		24/2	13/3	0.283
LVEF (%)		39.3 ± 8.68	35.3 ± 4.96	0.103
CHD event type	STEMI (%)	15 (57.7)	9 (56.3)	0.927
	NSTEMI (%)	11 (42.3)	7 (43.7)	
Hypertension		7 (26.9)	9 (56.3)	0.057
Diabetes mellitus		11 (42.3)	3 (18.8)	0.116
Dyslipidemia		19 (73.1)	8 (50.0)	0.130
Smoking status	Current	13 (50.0)	5 (31.3)	0.488
	Former	5 (19.2)	4 (25.0)	
	Never	8 (30.8)	7 (43.8)	
Family history		3 (11.5)	3 (2.3)	0.517
BMI (kg/m^2^)		25.3 ± 3.10	24.8 ± 2.22	0.576

Abbreviations: LVEF, left ventricular ejection fraction; CHD, coronary heart disease; STEMI, ST-elevation myocardial infarction; NSTEMI, non-ST-elevation myocardial infarction; BMI, body mass index.

**Table 2 healthcare-10-01849-t002:** Comparison of exercise capacity between CR participants and non-participants at initial and 3 months of ETT.

**Initial ETT**	**Cardiac Rehabilitation** **Participants (*N* = 26)**	**Cardiac Rehabilitation** **Non-Participants (*N* = 16)**	** *p* ** **-Value**
VO_2 peak_ (mL/kg/min)	23.07 ± 6.30	19.74 ± 4.94	0.079
VE _peak_ (L/min)	59.71 ± 18.85	52.53 ± 19.12	0.240
RER	1.06 ± 0.07	1.02 ± 0.09	0.146
Stage	4.81 ± 1.06	4.19 ± 1.11	0.078
METs	6.60 ± 1.80	5.64 ± 1.40	0.078
Exercise time (s)	753.38 ± 239.27	685.50 ± 165.71	0.326
SBP _max_ (mmHg)	167.88 ± 26.96	161.88 ± 27.54	0.491
DBP _max_ (mmHg)	76.96 ± 15.40	76.63 ± 13.78	0.943
HR _max_ (beat/min)	142.46 ± 18.92	136.13 ± 18.75	0.297
**ETT that Occurred** **After 3 Months**	**Cardiac Rehabilitation** **Participants (*N* = 26)**	**Cardiac Rehabilitation** **Non-Participants (*N* = 16)**	** *p* ** **-Value**
VO_2 peak_ (mL/kg/min)	26.24 ± 7.11	21.45 ± 6.40	0.033 *
VE _peak_ (L/min)	70.02 ± 23.02	56.15 ± 20.37	0.055
RER	1.09 ± 0.07	1.05 ± 0.08	0.197
Stage	4.88 ± 1.37	3.38 ± 1.45	0.002 *
METs	7.49 ± 2.06	6.13 ± 1.83	0.037 *
Exercise time (s)	768.08 ± 249.29	500.13 ± 235.49	<0.001 *
SBP _max_ (mmHg)	194.46 ± 123.80	167.19 ± 34.38	0.396
DBP _max_ (mmHg)	78.31 ± 10.91	80.69 ± 12.57	0.521
HR _max_ (beat/min)	150.15 ± 21.98	144.56 ± 17.85	0.396

Abbreviations: VO_2 peak_, peak oxygen consumption; VE _peak_, volume of air exchanged per minute; RER, respiratory exchange ratio; METs, metabolic equivalent of tasks; SBP _max_, maximal systolic blood pressure; DBP _max_, maximal diastolic blood pressure; HR _max_, maximal heart rate. * Denotes significant difference between the year group (* *p* < 0.05).

**Table 3 healthcare-10-01849-t003:** Comparison in exercise capacity changes in the initial ETT and of the ETT the ETT that occurred 3 months later between the CR participants and non-participants.

	**Cardiac Rehabilitation Participants (*N* = 26)**			
	Intial	3 months after	Δ	*p*-Value
VO_2 peak_ (mL/kg/min)	23.07 ± 6.30	26.24 ± 7.11	3.17 ± 4.01	<0.001 *
VE _peak_ (L/min)	59.71 ± 18.85	70.02 ± 23.02	10.31 ± 12.40	<0.001 *
RER	1.06 ± 0.07	1.09 ± 0.07	0.03 ± 0.07	0.064
Stage	4.81 ± 1.06	4.88 ± 1.37	0.08 ± 1.16	0.739
METs	6.60 ± 1.80	7.49 ± 2.06	0.89 ± 1.17	0.001 *
Exercise time (s)	753.38 ± 239.27	768.08 ± 249.29	14.69 ± 218.96	0.735
SBP _max_ (mmHg)	167.88 ± 26.96	194.46 ± 123.80	26.58 ± 119.59	0.268
DBP _max_ (mmHg)	76.96 ± 15.40	78.31 ± 10.91	1.35 ± 12.83	0.597
HR _max_ (beat/min)	142.46 ± 18.92	150.15 ± 21.98	7.69 ± 13.33	0.007 *
	**Cardiac Rehabilitation Non-Participants (*N* = 16)**			
	Intial	3 months after	Δ	*p*-Value
VO_2 peak_ (mL/kg/min)	19.74 ± 4.94	21.45 ± 6.40	1.71 ± 3.70	0.084
VE _peak_ (L/min)	52.53 ± 19.12	56.15 ± 20.37	3.62 ± 10.20	0.176
RER	1.02 ± 0.09	1.05 ± 0.08	0.03 ± 0.08	0.150
Stage	4.19 ± 1.11	3.38 ± 1.45	−0.81 ± 1.05	0.007 *
METs	5.64 ± 1.40	6.13 ± 1.83	0.49 ± 1.06	0.087
Exercise time (s)	685.50 ± 165.71	500.13 ± 235.49	−185.38 ± 186.12	0.001 *
SBP _max_ (mmHg)	161.88 ± 27.54	167.19 ± 34.38	5.31 ± 23.12	0.373
DBP _max_ (mmHg)	19.74 ± 4.94	21.45 ± 6.40	1.71 ± 3.70	0.084
HR _max_ (beat/min)	52.53 ± 19.12	56.15 ± 20.37	3.62 ± 10.20	0.176

Abbreviations: VO_2 peak_, peak oxygen consumption; VE _peak_, volume of air exchanged per minute; RER, respiratory exchange ratio; METs, metabolic equivalent of tasks; SBP _max_, maximal systolic blood pressure; DBP _max_, maximal diastolic blood pressure; HR _max_, maximal heart rate. * Denotes significant difference between the year group (* *p* < 0.05).

**Table 4 healthcare-10-01849-t004:** Changes of echocardiographic parameters between cardiac rehabilitation participants and non-participants.

Parameters	Cardiac Rehabilitation Participants (*N* = 26)				Cardiac Rehabilitation Non-Participants (*N* = 16)			
	Initial	3 months	Δ	*p* value	Initial	3 months	Δ	*p* value
EF (%)	37.76 ± 8.68	46.77 ± 11.82	9.01 ± 11.28	0.002 *	36.24 ± 4.03	43.64 ± 8.74	7.41 ± 8.21	0.005 *
LVESD (cm)	4.32 ± 0.88	4.04 ± 0.86	−0.28 ± 0.99	0.221	4.49 ± 0.81	4.22 ± 0.73	−0.27 ± 0.69	0.163
LVEDD (cm)	5.42 ± 0.62	5.45 ± 0.77	−0.03 ± 0.84	0.875	5.59 ± 0.66	5.54 ± 0.54	−0.05 ± 0.46	0.718

Abbreviations: EF, ejection fraction; LVESD, left ventricular end-systolic diameters; LVEDD, left ventricular end-diastolic diameters. * Denotes significant difference between the year group (* *p* < 0.05).

## Data Availability

The datasets generated and/or analysed during the current study are not publicly available but are available from the corresponding author on reasonable request.

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
