# Peer review of "Effects of Cardiac Rehabilitation in Cardiopulmonary Fitness with High-Risk Myocardial Infarction"

_healthcare, 2022, doi:10.3390/healthcare10101849_

Round 1

Reviewer 1 Report

Abstract:

“There are few 9 studies have been conducted in myocardial infarction (MI) patients with reduced heart function so 10 called the high risk group.” Consider changing the term so-call high risk group to HFrEF being a high risk condition

In the abstract, please mention the baseline characteristics of the patents included.

Introduction:

-There have been many RCTs that have evaluated and proved the efficacy of cardiac rehab. Please elaborated in the introduction about what does this study add to the literature?

-Consider condensing the introduction section to ½ page and restrict to 2 paragraphs by focusing on the most important background information and conveying what the study main aims and objectives were.

Methods:

Are the physician investigators mentioned in methods, co-authors on this paper? If yes, I would suggest to add their initials rather than mentioning just “Physician investigators”

Was there any missing data in the dataset and how was it handled?

Did the authors follow the Strobe reporting guidelines?

How did the authors calculate the sample size for their study population?

Results:

Consider not repeating results in the tables again in the text.

Discussions/Conclusions:

Please elaborate on what are the clinical implications of your study findings and what does the study add compared to previous RCTs on the topic.

Please acknowledge important limitations such as the study findings are only hypothesis generating. The lack of external validity of the study. The small sample is also a major limitation that must be discussed and acknowledged.

Author Response

Thank you for the paper review.

The points have been revised as you commented.

Line numbers were written based on the revised paper, using ‘the track changes mode in MS Word’.

Reviewer 2 Report

I think it’s an interesting work. I would recommend it if the following issues are handled properly.

1. Provide the basis of exercise tolerance tests. Do you follow which protocols to design and carry out these exercise tolerance tests?

2. Display your echocardiography figures.

3. Explain the indicators of the metabolic equivalent of tasks (METs) values in detail.

4. Provide one schematic to exhibit the protocols and evaluations of cardiac rehabilitation in this research.

Author Response

(The authors gave the same response as above.)
